# The Association between Music and Language in Children: A State-of-the-Art Review

**DOI:** 10.3390/children10050801

**Published:** 2023-04-28

**Authors:** Maria Chiara Pino, Marco Giancola, Simonetta D’Amico

**Affiliations:** Department of Biotechnological and Applied Clinical Sciences, University of L’Aquila, 67100 L’Aquila, Italy; mariachiara.pino@univaq.it (M.C.P.); marco.giancola@univaq.it (M.G.)

**Keywords:** child, communication, infants, language development, music, review

## Abstract

Music and language are two complex systems that specifically characterize the human communication toolkit. There has been a heated debate in the literature on whether music was an evolutionary precursor to language or a byproduct of cognitive faculties that developed to support language. The present review of existing literature about the relationship between music and language highlights that music plays a critical role in language development in early life. Our findings revealed that musical properties, such as rhythm and melody, could affect language acquisition in semantic processing and grammar, including syntactic aspects and phonological awareness. Overall, the results of the current review shed further light on the complex mechanisms involving the music-language link, highlighting that music plays a central role in the comprehension of language development from the early stages of life.

## 1. Introduction

Due to their similarities, the music-language link has received particular consideration among infants and preschoolers [1]. Indeed, music and language are two hierarchical systems [2] in which smaller and separate units (e.g., phonemes and notes) are combined into higher-order structures by specific rules (e.g., words, sentences, and musical compositions). Both systems are characterized by sequences of melodic and rhythmic patterns, relying on melody and rhythm in music and prosody in language [1]. According to François & Schön [3], both systems are based on different perceptual and cognitive processes such as sound identification and categorization and memory storage and retrieval. Finally, both systems represent the most powerful forms of human communication, with the same stimuli being perceived as either language or music, depending on the listener’s interpretation [4].

The acquisition of both language and music has been argued to lie in a general principle: learners extract statistical patterns and regularities in the sound environment [5]. The implicit perceptual mechanism responsible for the incidental acquisition of structure in one’s environment has been termed statistical learning, and it is thought to be present before birth and throughout life. It has a crucial role in the development of many abilities, including learning the sound structure of the language and the music of the culture. Brown [6] suggested that both speech and musical phrases are “melodorhythmic” structures in which melody and rhythm are derived from three sources: (1) the acoustic properties of the fundamental units; (2) the sequential arrangement of such units in each phrase; and (3) expressive phrasing mechanisms that modulate the basic acoustic properties of the phrase for expressive emphasis and intention.

Some theoretical approaches support the hypothesis that language and music had a common precursor [6,7], known as protolanguage. The protolanguage has defined a language without syntax; it refers to either a holophrastic or arbitrarily concatenated language [6]. Specifically, according to Mithen [7], there are two main approaches to the evolution of language regarding the nature of protolanguage. The first approach is known as “compositional” and is associated with the theory of Bickerton and Jackendoff, suggesting that words came before grammar and that it is the evolution of syntax that differentiates the vocal communication system of Homo sapiens from all of those that went before. The second alternative approach, described by Wray and Arbib, suggests that pre-modern communication was constituted by ‘holistic’ phrases, each of which had a unique meaning and which could not be broken down into meaningful constituent parts [7]. In line with the holistic approach, Mithen [7] supports the idea that the phrases also make extensive use of variation in pitch, rhythm, and melody to communicate information, express emotion, and induce emotion in other individuals. As such, both language and music have a common origin in a communication system defined by Mithen [7] as “Hmmmmm” and by Brown [6] as “musilanguage”, that had the following characteristics: holistic, manipulative, multi-modal, musical, and mimetic. The philosopher Rousseau [8] argued that ancestral humans would have used a “protomusilanguage” and that people would have communicated by singing [9]. According to Brant et al. [4], language is a type of music where referential speech is involved in a musical structure. This aligns with Darwin’s hypothesis of similarities between language and music, in which human infants and young children were born with variable musical capabilities while language evolved from an early musical communicative system [10,11]. Interestingly, a form of communication in which music and language overlap seems to be infant-directed (ID) speech, also known as “motherese” [12,13]. Parents’ use of ID speech containing exaggerated prosodic properties is important in facilitating speech perception and language acquisition. This latter is characterized by a higher pitch, a slower tempo, the repetition of shorter sequences and sustained pauses, and amplified rhythmic and melodic patterns that are specific to one’s native language [12,14,15]. All these characteristics of ID speech support infants’ vocal development and modulate the interaction between parent and child [15,16,17,18]. Kotilahti et al. [19] demonstrated that newborns show largely overlapping brain activations between ID speech and instrumental music.

Growing evidence suggests that several aspects of language and music processing in humans involve similar change detection processes, from acoustic changes to the statistical structure of a sequence of sounds [2]. François and Schön [3] argued that the detection of a change, strongly linked to the predictive abilities of the auditory system, can be considered a biological process involved in the statistical learning of music and language. Concerning the neural processes underlying music and language, research on patients with cerebral damage supported the idea that these two systems shared resources. For example, Broca’s aphasia is characterized by failure to process musical syntax (for details, see [18]). Despite these similarities, some authors found opposite hemispheric dominance for speech over singing in the left temporal lobe, whereas singing over speech occurs in the right temporal lobe [20]. Furthermore, dissociations between music and language were found in patients with impaired processing of harmonic relations with preserved linguistic syntactic processing [1,21] or in patients with impaired grammatical processing of language with musical syntax intact [1,22]. Overall, these findings stressed that the dissociations between music and language depend on structures and functions, which make them two distinct and separate systems with unique features [23]. In this vein, Slevc and Patel [24] identified three critical differences between music and language regarding their meaning. First, the meaning evoked by music is far less specific than the meaning evoked by language. Specifically, units of a language denote specific semantic concepts, whereas units of music pick out semantic concepts at a much coarser grain. Second, linguistic semantics is compositional, unlike musical semantics. This implies that, whereas words can be combined in lawful ways in order to give rise to more complex meanings, units of music cannot be combined to convey propositions. Third, semantics exists for communicative reasons compared to music, which might better be conceived of as a form of expression rather than communication.

Beyond similarities or differences, other factors can affect the interactions between language and music. Indeed, they mutually rely on genetic, cognitive (e.g., intelligence), extra-cognitive (e.g., personality traits) and environmental factors (e.g., socioeconomic status), which provide high interindividual variability in their acquisition [25,26,27].

Nevertheless, there is agreement that music and language show a deep early entanglement as well as largely similar developmental trajectories in the early years of life.

Based on these premises, the present review aims to shed light on the relationship between musical and linguistic aspects in infants and school-aged children by reviewing existing literature and investigating if music and language are associated in the early stages of life.

## 2. Method

### 2.1. Search Strategy and Inclusion and Exclusion Criteria

An extensive literature search has been carried out for the present review using three databases (Pubmed, Scopus, and Web of Science), and the following string of keywords was used: music AND language AND NOT disease AND NOT bilingual AND NOT disorder AND NOT brain AND NOT second language AND NOT adults. Then, the articles obtained were pruned according to different inclusion/exclusion criteria, as follows:(a)only articles written in English and published in peer-reviewed journals; no conference articles, reviews, meta-analyses, case reports, letters to the editor, or book chapters;(b)There are no articles on the association between music and language in children with clinical conditions or disorders. Articles on neurodevelopmental disorders such as specific language disorders, cognitive delay, deafness, autism spectrum disorders, neurological deficits, pervasive developmental disorders, traumatic brain injuries, primary disorders (sensory, neurological, or psychiatric), dysphonia, dysarthria, dysrhythmias, stuttering, specific speech articulation disorder, or dyslexia were excluded. This allows for exclusively addressing the relationship between music and language in typical development.(c)no articles on music therapy or music rehabilitation. This exclusion criterion is strongly associated with the criterion “b” because, in this specific review, we were not interested in considering the role of music in the habilitative or rehabilitative context of specific disorders;(d)There are no articles investigating the role of musical training in language development. Studies that included music training were excluded because we believe that this topic would require a dedicated review involving structured and unstructured training types with different features, such as individual or group level, professional figures involved (teachers, parents, musicians or music therapists, other professionals), duration and frequency of the intervention, and settings (home, clinics, community, school);

There are no articles addressing the influence of musical experience on second language (L2) acquisition. Studies that included L2 and/or bilingual children were excluded because these issues would have introduced several new variables to be accounted for, such as the age of second language acquisition, levels of proficiency requested, and the number of acquired languages that would deserve a dedicated review.

During the study selection process, literature was evaluated by the authors, considering duplicates, followed by a screening of titles and abstracts. Afterward, results were screened in full text if considered eligible based on inclusion and exclusion criteria, and for each included research study, the following data were extracted: (1) research design and sample characteristics, including size, age, and gender; (2) type of music measures; (3) type of language measures; and (4) findings

Experts and practitioners in the field indicated further potentially relevant studies. Titles and abstracts were screened by two independent reviewers to exclude irrelevant studies according to our inclusion/exclusion criteria. Potentially eligible studies were extracted in full text and screened for eligibility. Disagreements were subsequently resolved.

Through bibliographic searches, 2120 studies were identified. Three articles were also included as additional records identified through other sources. The research articles obtained as output were collected in a unique electronic sheet, reporting the related authors’ list, title, year of publication, and source. The duplicates (n = 611) have been removed, and two independent reviewers evaluated the remaining 1513 articles based on the title and abstract. Nineteen studies were retrieved in full text for a more detailed evaluation, and seven articles were excluded due to being off-topic. Finally, 12 papers were included in our review (Table 1).

### 2.2. Research Design and Sample Characteristics

As shown in Table 1, the 12 selected articles in this review have been published from 2015 to 2023. Ten articles are cross-sectional studies that used correlations or/and regression analyses to investigate the associations between some musical properties and some linguistic competencies. Two articles are longitudinal studies that have collected data from at least two-time points from the same participants over time. Only one of the 12 selected articles is an EEG study [28], which reveals the electrophysiological brain dynamics during statistical learning from continuous flows of sung and spoken syllabic sequences in newborns.

Regarding sample characteristics, the 12 selected studies considered the following age groups: eight articles included school-aged children with an age range of approximately 5 to 11 years; only one article included preschool children; and finally, three studies involved infants. Regarding language, nine articles involved children who were monolingual or exposed to one language. The language most frequently involved is English, followed by German; in the three articles, the participants’ speech is Italian, Spanish or Catalan, and French. In the article by Dolscheid et al. [29], participants were half Dutch natives and half Turkish children. Some participants were bilingual or exposed to an L2 in the three articles. First, in Swaminathan & Schellenberg’s study [30], 47 children of 91 participants were bilingual (or multilingual), and the authors kept this variable constant in the statistical analysis. Second, in Politimou et al.’s study [1], 12 children out of a total of 40 participants were bilingual but had English as their first language. In the preliminary analysis, the authors [1] showed no significant difference between monolingual and bilingual children. Third, in the article of Franco et al. [31], 5 of 36 infants were also exposed to an L2, but not bilingual infants.

Regarding the gender variable, all 12 selected articles showed a good sample distribution, as about half of the participants were female, with the other half being male.

### 2.3. Music Measurement

Regarding musical measures, six of the 12 selected articles used standardized tests, and four articles collected data on the home musical environment [1,15,30,31,32,33]. Specifically, Franco et al. [31] and Politimou et al. [1] used the Musical Experience in the Family Questionnaire (MEF) [34]. Cohdes et al. [35] adopted the home musical environment scale (HOMES) [36], whereas Papadimitriou et al. [15] employed the Music@Home-Infant [37] and two subscales from the Goldsmiths Musical Sophistication Index (Gold-MS, [38]), i.e., the Musical Training subscale and the Active Engagement with Music subscale. Gold-MSI [38] was also used by Politimou et al. [1].

To evaluate musical rhythm perception, Gordon et al. [32] and Nitin et al. [33] used the children’s version of the beat-based advantage (BBA) assessment. Gordon et al. [32] also used an additional rhythm measure: the rhythm section of the primary measures of music audition (PMMA; [39]). See Appendix A for details about the standardized tests used in the articles selected in this review to assess musical abilities.

Swaminathan and Schellenberg [30] collected the musical expertise variable according to parent reports without using a specific standardized test to collect this information. This variable involved the average number of months of music lessons their children took privately or in school. In addition, in this study, the authors [30] used three sub-tests of the Montreal Battery of Evaluation of Musical Abilities (MBEMA; [40]), which they adapted as a video game.

Seven articles out of 12 selected articles used experimental paradigms constructed ad hoc by the researchers to assess different musical aspects, such as space-pitch [1,29], melody [1,28], synchronization [1], tunes [31], rhythm [41,42,43], tempo [1], and textual sound sequences [42].

Only one selected article employed the experimental paradigm to collect the physiological responses during musical and/or linguistic stimulus [28]. Specifically, in this study, EEG was recorded from 16 scalp electrodes of 28 neonates located at standard positions, and infants were submitted to an experimental paradigm with two conditions (flat contour and melodically enriched conditions). Two languages were built using a set of 24 synthetic syllables that were combined to give rise to two sets of four tri-syllabic pseudo-words. In the flat contour condition, all the syllables were spoken on the same pitch, resulting in a monotonous stream of syllables in which the only cue to word segmentation was the transitional probabilities between adjacent syllables. In the melodically enriched condition and for each language, each of the 12 syllables was associated with a distinct tone. Therefore, each word had a unique melodic contour. Infants were lying asleep in their cribs while presented with both conditions’ auditory streams [28]. All infants were presented with a flat contour condition, followed by the melodically enriched condition.

### 2.4. Language Measurement

Regarding language measurement, eight of the 12 selected articles used standardized tests to evaluate linguistic abilities [1,15,28,30,31,32,33,41]. Specifically, Swaminathan and Schellenberg [30] employed the Test for Reception of Grammar, Version 2 (TROG; [44]) to measure knowledge of language grammar. Chern et al. [41] instead selected the participants based on the score obtained by the Test of Language Development (>85; [45]). Both articles [30,41] used these specific language tests not as an outcome variable of the study but as a screening variable for good language skills. Swaminathan and Schellenberg [30] adopted a modified version of the AXB discrimination task [46] to evaluate speech perception.

Gordon et al. [32] and Nitin et al. [33] used the Structured Photographic Expressive Language Test (SPELT-3; [47,48]) to evaluate the children’s expressive grammatical abilities. In addition, Gordon et al. [32], in their study, adopted the Comprehensive Test of Phonological Processing (CTOPP; [49]) to measure phonological ability. François et al. [28] and Franco et al. [31] have used the Spanish [50] and Italian versions [51] of the MacArthur-Bates Communicative Development Inventory [52], respectively. François et al. [28] also used the language subscale scores from the Bayley Scales of Infant Development (BSID-III [53]). In Politimou and colleagues’ article [1], language grammar has been assessed with the Language Structure Index (LSI) from the Clinical Evaluation of Language Fundamentals, Preschool-2 (CELF-Preschool-2; [54]). Two tests of rhyme and alliteration awareness—the Phonological Oddity-Rhyme and the Phonological Oddity-Alliteration task [55]—replaced the rest of the phonological awareness subtests of the CELF-Preschool-2 (Rhyme perception and Rhyme generation) because, during pilot testing, these were deemed inappropriate for the younger as well as some of the older children. Finally, Papadimitriou et al. [15], used the UK-Communicative Development Inventory (UK-CDI) to assess infant language [15,56]. Appendix B provides details about the standardized tests used in the articles selected for this review to assess linguistic abilities.

**Table 1 children-10-00801-t001:** The studies included in the review.

Reference	Research Design	Sample Characteristics	Music Measurement	Language Measurement	Main Findings
Politimou et al. [1]	Cross-sectional	Study 1 40 (21 boys) children with a mean age of 4 years (SD = 4.7 months) participated in the study. 28 children were native speakers of English, and 12 children were bilingual but had English as their first language Study 2 34 parents of children who participated in Study 1 Mage = 36.4	Music perception: Pitch perception task Melody perception task Melody discrimination task Tempo perception task Music Production: Song Production. Sing along to the recording of “Twinkle Twinkle Little Star”. Synchronized tapping task. Tap along metronome clicks Informal Music Experience at Home: Musical Experience in the Family Questionnaire [34]; Goldsmiths Musical Sophistication Index Gold-MSI [38].	Language grammar: The Language Structure Index (LSI) of the Clinical Evaluation of Language Fundamentals–Preschool-2 (CELF-Preschool-2, [54]). Phonological Awareness: The Phonological Oddity. Alliteration task.	An increase in rhythm perception performance and lower synchronization errors were associated with better phonological awareness. In addition, better perceptual processing of melodies was associated with higher grammar scores. Finally, informal musical experience at home contributes to grammar development and mediates the effect of musical skills on both phonological awareness and language grammar. Results showed that the informal musical in the domestic environment, such as parental singing, has a direct impact on the development of gestural conversation in children and significantly predicted their word comprehension
Papadimitriou et al. [15]	Cross-sectional	64 infants (range age from 8.5 to 18 months; 37 F, 27 M). English is the only language spoken at home.	Informal Music Experience at Home: The Music@Home-Infant [37] Two subscales from the Gold-MSI [38]: Musical Training and Active Engagement subscales.	Comprehension, production, and gestural communication: UK-Communicative Development Inventory: Words and Gestures (UK-CDI [56])	Results showed that the home musical environment has a direct implication for generating a gestural conversation. For infants below 12 months, parental singing and overall home musical environment scores significantly predicted word comprehension.
François et al. [28]	Longitudinal design (Electrophysiological study)	38 healthy neonates with a mean age at testing = 2.8 days (SD = 0.9 days, 14 male). Parents of infants involved in the study spoke either Spanish or Catalan .	Ad-hoc laboratory test (melodically enrich condition vs. flat contour condition) with melodic stimuli aimed to assess melodic competencies.	Expressive vocabulary: Spanish version of the MacArthur-Bates Communicative Development Inventory [50,53] Bayley Scales of Infant Development (BSID-III [53])	The results showed that early neural individual differences in prosodic speech processing might be a precursor of later language abilities and could have a significant role in the development of language in infants.
Dolscheid et al. [29]	Cross-sectional	80 Dutch native children and 80 Turkish children. Sample was divided into four age groups with 40 children per group: 5-, 7-, 9-, and 11-year-olds. For each group, 20 F and 20 M.	Non-linguistic height-pitch and thickness-pitch: Nonlinguistic space-pitch association tasks Nonlinguistic space-pitch conflict task	Height and thickness pitch terminology: Linguistic space-pitch comprehension tasks	The results suggest that thickness-pitch associations are acquired in similar ways by children from different cultures, but the acquisition of height-pitch associations is more susceptible to linguistic input. Overall, despite cross-cultural stability in some components, there is variation in how children come to represent musical pitch, one of the building blocks of music.
Swaminathan & Schellenber [30]	Cross-sectional	91 TD children (46 M;45 F) between 6 and 9 years of age (mean = 7.84 years, SD = 1.19). 44 native speakers of English and 47 bilinguals (English and French)	Melody rhythm discrimination: Montreal Battery of Evaluation of Musical Abilities (MBEMA; [40]), Children’s musical expertise: Musical expertise, according to parent reports	Speech perception: A modified version of the AXB discrimination task [46] Grammar comprehension: Test for Reception of Grammar–Version 2 (TROG [44]).	After controlling for bilingualism, rhythm discrimination was a better predictor of language skills than melody discrimination.
Franco et al. [31]	Longitudinal design	36 Italian Children available at T1 (50% F), with 5 infants also being exposed to an L2. Age = 6 months. 26 Children available at T2 (46% F). Age = 14 months	Music experience: Ad-hoc laboratory test with three pairs of tunes (approx. 1 min each in duration, 68.5 s on average). Informal Music Experience at Home: Musical Experience in the Family Questionnaire (MEF [34])	Expressive vocabulary: The Italian version of the MacArthur-Bates Communicative Development Inventory [51,52]	Self-reported high levels of parental singing predicted children’s language outcomes in the second year.
Gordon et al. [32]	Cross-sectional	25 English children (13M;12F), aged 5 years and 11 months to 7 years and 1 month (M = 6 years and 6 months, SD = 4 months)	Rhythm discrimination: The beat-based advantage (BBA [32]) The Primary Measures of Music Audiation (PMMA [39])	Expressive grammatical abilities: Structured Photographic Expressive Language Test (SPELT-3; [48]) Phonological awareness: Comprehensive Test of Phonological Processing (CTOPP; [49])	Children with higher phonological awareness scores were better at discriminating complex rhythms than children with lower scores. This study is the first to show a relationship between rhythm perception skills and morpho-syntactic production in children with typical language development.
Nitin et al. [33]	Cross-sectional	132 children (age range = from 5 years and 1 month to 8 years and 1 month; mean age = 6.5 years, SD = 10 months, 76 females). English is the primary language spoken at home	Rhythm discrimination: Musical rhythm perception (BBA [32])	Expressive grammatical abilities: Structured Photographic Expressive Language Test-3 (SPELT-3 [48])	Results showed that music rhythm perception predicted prosody perception, expressive grammar, and complex syntax.
Cohrdes et al. [35]	Cross-sectional	44 German children aged 5 to 7 years (53.50% female) and a control group of 20 young adults aged 20 to 30 (45% female).	Children’s musical expertise:Home musical environment scale (HOMES) [36] Music competencies Melody repetition Tonal discrimination Sound discrimination Rhythm repetition Harmonic progression Emotion recognition in tonal sequences Synchronization	Language competencies: Phonemic discrimination; Word discrimination Phonological awareness; Syntactic integration Emotion recognition in spoken phrases Narrative Comprehension	Results indicate that language and music skills are significantly interrelated. Additionally, in line with a hierarchical model of skill acquisition, performance on lower levels was predictive of performance on higher levels in both the musical and linguistic domains.
Chern et al. [41]	Cross-sectional	16 English children aged 5 years and 6 months to 8 years and 7 months (M = 6 years and 5 months, SD = 11 months; 9M, 7F)	Rhythmic priming effect: Rhythmic priming effect paradigm Rhythmic priming effect: Grammaticality judgment task with musical stimuli	Language competencies: Test of Language Development [45]	The results showed the Rhythmic Priming Effect causes enhanced grammar task performance in children with typical language, suggesting a robust benefit of rhythmic listening and highlighting its possible relevance for language development.
Canette et al. [42]	Cross-sectional	30 native French-speaking children (50% F): 16 performed the syntax task (8 M; average chronological age = 8 years 0 months, SD = 6 months, range = 7 years 2 months to 8 years 11 months), and 14 performed the semantic task (7 M; average chronological age = 7 years 11 months, SD = 4 months, range = 7 years 6 months to 8 years 5 months). 16 other children participated in a follow-up of the syntax task (7 M; average chronological age = 7 years 3 months, SD = 5 months, range = 6 years 10 months to 8 years 7 months).	Music experience: Regular rhythmic sequences Textural sound sequences	Language competencies: Syntax task Semantic evocation task	The results revealed that rhythmic and textural musical sequences influence syntax and semantic processing differently. For grammaticality judgments, children’s performance was better after regular rhythmic sequences than after textural sound sequences. In the semantic evocation task, children produced more numerous and varied concepts after textural sound sequences than after regular rhythmic sequences.
Lee et al. [43]	Cross-sectional	Study 1 N = 68 children (35 girls) Mage = 11.30 years SD: 2.7 years Study 2 N = 96 children (56 girls) Mage = 11.10 years, SD 2.7 years Native English-speaking children	Rhythm competencies: Rhythm test	Syntax competencies: Grammar test	As children improved on the rhythm task, they also improved on the grammar task.

Six articles used experimental paradigms to assess different linguistic aspects, such as linguistic space-pitch comprehension ability [29], phonemic contrasts [30], phonological awareness [32], grammar [41,42,43], and syntax and semantic aspects [42].

According to the music measurement session, one article used physiological responses to investigate the neural mechanisms involved in the relationship between language and music through ad-hoc laboratory tests without melodic stimuli (flat condition, see [28]). 

Finally, one article [35] used some items from different tests to evaluate several language competencies, such as phonemic discrimination, word discrimination, phonological awareness, and syntactic integration. For example, the authors [35] employed three subscales of a detection dyslexia test in the German language called Das Bielefelder Screening (BISC, [57]) to evaluate phonological awareness; additionally, they used a German test of speech development (SETK3-5; [58]).to assess prosodic repetition.

## 3. Findings

All 12 selected articles support the association between music and language abilities. Specifically, these articles advance the hypothesis that musical skills predict children’s performance on language tasks. The 12 papers provide evidence that rhythm and melody predict language acquisition, including semantic and syntactic aspects as well as phonological awareness. Additionally, three of the 12 articles highlight the value of informal musical experience in a domestic environment for children’s language development.

Regarding the associations between rhythm and language, literature agrees that musical rhythm and language, in terms of grammar and phonological awareness, share a high number of intrinsically related characteristics [41]. Specifically, for the rhythm-grammar link, Swaminathan and Schellenberg [30] affirmed that rhythm discrimination was a better predictor of performance on the tests of receptive grammar. Similarly, Nitin et al. [33], Chern et al. [41], and Lee et al. [43] showed a robust effect of music rhythm perception on grammatical task performance.

The association between rhythm and phonological awareness was explained by Gordon et al. [32], where the authors showed that children with higher phonological awareness scores were better able to discriminate complex rhythms than children with lower scores. This relation between rhythm and phonological awareness was confirmed in Politimou and colleagues’ study [1], in which an increase in rhythm perception performance and lower synchronization error were associated with better phonological awareness.

Canette et al. [42] introduced two new components within the relationship between music and language, namely texture and musical and semantic processing. Specifically, the authors [42] clarified that rhythm seems particularly relevant to enhance syntax processing, whereas textural musical sequences seem important to promote semantic activation.

Furthermore, research suggests that specific musical features, such as melody, play an essential role in language development. Francois et al. [28] provided converging neural and computational evidence of the early benefit of melodies for language acquisition. Indeed, the neonates’ brain responses to sung streams predicted expressive vocabulary at 18 months. Their findings suggest that early neural individual differences in prosodic speech processing might be a good indicator of later language outcomes and could be considered a relevant factor in developing infants’ language skills. Politimou and colleagues’ study [1] focused on the association between melody and grammar development. Their results showed that 3- and 4-year-old children could extract regularities from musical environments by internalizing melodic and harmonic structures. Similarly, between 3 and 4 years of age, children internalize the structures of their native grammar. Yet, the results of Cohrdes et al.’s study [35] demonstrated that the ability to discriminate melody in children with an age range of 5 to 7 years was related to the ability to process emotional prosody in linguistic sentences, supporting the hypothesis that basic auditory skill strengthens more fine-grained linguistic aspects.

Finally, three selected articles emphasized the influences of infant-directed (ID) speech on later language development [1], mainly in terms of gestural interaction, word comprehension [15], and communication skills [31]. This communication form is more attractive for infants and allows them to easily learn aspects of phonetic perception and word learning [59,60]. Specifically, Franco et al. [31] found that self-reported high levels of parental singing predicted children’s language outcomes in the second year, while Papadimitriou et al. [15] showed that the home musical environment significantly predicted gesture development. The gesture represents a critical component of children’s communication skills development and is one of the precursors of intentionality [61]. Gesture plays a valuable role in several functions, including communication, compensation for language absence, and transition to spoken language. In detail, Papadimitriou et al. [15] found that parental singing and the overall home musical environment score significantly predicted word comprehension in infants under 12 months. Interestingly, the relationship between music in a home environment and language development was not seen in older infants (12–18 months). This could be explained by the fact that, for infants 12 months and older, there is typically greater variability in terms of environmental input (linguistic, social, and musical). Both of these findings [15,31] demonstrate that an enriched musical environment in infancy can promote the development of communication and complex language skills.

## 4. Discussion

Music and language represent two critical communicative systems that characterize human beings [1,2,4]. They involve the combination of an elementary set of sounds ordered in time according to rules, allowing the perception and production of complex and unlimited utterances or musical phrases [3]. Research emphasizes that musical activities are predictive of language development. For this reason, children are exposed at an increasingly early age to musical activities at home and in more structured environments [31,62]. Since music and language are composed of various distinct components, it is crucial to disentangle and understand the specific elements of one domain that are associated with components of the other domain. Therefore, this review sought to deepen the understanding of which musical properties affect the linguistic components according to the development approach.

The selected articles showed that rhythm is the musical component that predicts infants’ language development, mainly in expressive and receptive language. Rhythm can be considered the structured arrangement of successive sound events over time, a primary parameter of musical structure [63,64]. A growing body of research in language development suggests that rhythm components influence early linguistic production in children [65,66,67,68,69]. Particularly, this evidence showed that infants use rhythmic cues to identify word-level units in speech [70]. Infants show responsiveness and sensibility to the rhythmic linguistic components and are able to distinguish among languages based on their rhythmic structures [4,71]. Several studies reported evidence about the rhythmic classes used to classify the language [72,73,74,75]. Specifically, literature agrees with a three-way classification of languages according to their predominant rhythmic structure [76,77]: Romance languages, such as Italian or Spanish, have a rhythm based on the syllable; most Germanic languages, such as English, Dutch, and German, have a rhythm based on the stress unit, while languages such as Japanese have a mora-based rhythm. Nazzi et al. [73] showed that at five months, infants may prefer any language with the same rhythmic structure as their native language (for example, German and English). On the contrary, 5-month-old infants discriminate pairs of languages from different rhythmic classes (e.g., English vs. Japanese; Italian vs. Japanese). This means that infants may not prefer their native language but rather the rhythmic characteristics of that language [4,78]. Moreover, infants’ early attention to rhythm (e.g., [75,79]) suggests that they are absorbing the sonic structure of their native language in the same way that we listen to music [4]. Moreover, it was demonstrated that at 12 months, the infants might already show a musical enculturation form for rhythm. Broadly in support of these findings, it has been suggested that rhythmic skills such as rhythm discrimination [80] and rhythm production may develop earlier than melodic abilities, which instead seem to emerge approximately every 3 and 4 years [1,81]. Interestingly, the rhythmic properties of music and language share some notable features: music and language are grouped into phrases marked by pauses and differences in tone height and duration of beats and syllables [82]. Given the many parallels, it has often been proposed that shared cognitive or perceptual mechanisms are active in the acquisition (e.g., [83]) and/or processing [84] of music and language. Rhythm production skills [30,41,42,43] and pitch [29] turned out to be predictive of expressive grammatical abilities in schoolchildren [30,32]. Overall, this evidence suggests that if such shared mechanisms are active, then music stimulation should reasonably facilitate language development.

During a child’s development, beyond rhythm, melody represents an additional musical component affecting language, which becomes more complex. Melody is described as patterns of pitched sounds unfolding over time, following cultural conventions and constraints [64]. Politimou et al. [1] have provided, for the first time in preschoolers or other developmental groups, evidence of an association between melody perception and language grammar. These results strongly suggest that similar auditory perceptual mechanisms may be responsible for both melody perception and language grammar, at least at this stage in development [1]. Thus, according to Politimou et al. [1], for preschoolers, rhythm and melody in different ways predict phonological awareness and language grammar, respectively.

The strong link between music and language through different stages of development, from infancy to schoolchildren, suggests that these two systems share common competencies and neural resources. According to Cohrdes et al. [35], there are similarities in the syntactic processing between musical and linguistic domains. Their results suggest that the competency to integrate smaller units into a higher-level syntactic system builds upon the abilities to discriminate as well as reproduce smaller units such as phonemes and sounds, words and tonal phrases, and syllables and rhythmic phrases. Generally, this indicates that competencies on higher levels might build upon different competencies on lower levels, and this characteristic is common for both domains. This finding is in line with the theories according to which musical and linguistic development support a step-by-step acquisition of skills in both modalities [1,35,85,86]. Such distinct abilities rely on common learning mechanisms that are partly dissociable from domain-specific aspects. Regarding shared neural resources, several behavioral and neuroscientific studies both in children and adults have supported the idea of shared online processing [22,87,88,89,90,91,92,93], such as the bilateral frontal-temporal network [4,82]. Specifically, Schön et al. [92] revealed bilateral involvement of the middle and superior temporal gyri and the inferior and middle frontal gyri while listening to spoken words, sung words, and “vocalize”, i.e., singing melodies without words [92]. Thus, it has been hypothesized that music can have a privileged status in the infant’s brain, enabling them to acquire and strengthen linguistic abilities [4]. Therefore, future research should identify the mechanisms underlying the music and language link along the developmental trajectory [1].

Another interesting aspect that emerges from selected studies is the primary role of ID speech on language development. Particularly, evidence highlights that ID speech predicts children’s language abilities in the later years, about 2 years [31], or in 3- and 4-year-old children for more complex language skills [1], such as the grammatical structure of language. Additionally, these studies suggest that higher levels of home engagement with singing, music making, and greater exposure to music can serve as scaffolding for acquiring verbal skills, greatly extending previous suggestions [1].

The current review provides evidence that music plays a critical role in language development in early life. This research has at least three implications worth mentioning. First, our review suggests that infants showed sensitivity to the rhythmic components of language, and this musical component turned out to be predictive for the development of expressive language components such as grammatical ability and phonological awareness. Second, the results of the articles selected showed that melody ability seems to be associated with complex and refined linguistic abilities, such as the processing of emotional prosody in linguistic phrases. This result suggests that music and language show a deep entanglement as well as largely similar developmental trajectories since the early years of life. Third, our review highlights the effect of the home’s informal musical environment, such as ID speech, lullabies, and play songs, on later language development.

Despite the findings, note that articles investigating the effect of musical interventions such as music training, music therapy, and music rehabilitation were excluded. Although previous research consistently addressed this topic, our decision is motivated by the complexity of the music interventions that involve too many additional variables to be accounted for, such as approaches and modalities of music, individual or group levels, professional figures involved, frequency and duration of the interventions, and settings (e.g., at home or school). Although we know that our decision to exclude articles focusing on music interventions might have limited the scope of the review, our results provide reasonable evidence that language development benefits from musical skills. Thus, music has a central role in the comprehension of language development in the early stages of life.

An important consideration that deserves to be mentioned is the poor representation of non-western languages. It introduces a bias towards these languages and limits the generalizability of the findings on the relationship between music and language in other languages. To our knowledge, no studies have directly investigated the association between music and language according to our inclusion and exclusion criteria, and this represents a significant limitation of research in this specific field.

Future research should investigate the specific role of music interventions on language development, considering the music-language link, as well as other factors that can affect the interactions between language and music, such as cognitive skills, personality traits, and environmental dimensions. Accordingly, future investigations of associations between musical and linguistic abilities should account for inter-individual variability.

## 5. Conclusions

The present review sought to provide an overview of the existing literature on the relationship between music and language at the first stage of life. Although these systems show some differences in neural mechanisms, the current review supports the view that they share some characteristics that cannot be ignored. Overall, results highlight that music components, such as rhythm and melody perception as well as synchronization and informal experience of music in a domestic environment, play a main role in language development, mainly in terms of phonological awareness, grammar, prosody, and comprehension, since the early stages of life. This literature review aims to create the basis for future investigations. In particular, the upcoming challenge is to understand the effects of musical interventions both in children with typical development and in a clinical setting.

## Data Availability

Not appliable.

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
