# Peer review of "The Association between Music and Language in Children: A State-of-the-Art Review"

_children, 2023, doi:10.3390/children10050801_

Round 1

Reviewer 1 Report

This paper provides a limited review of the relationship between music and language in children, focusing on their similarities and differences, the processes involved in their acquisition, and touches vaguely the neural mechanisms underlying their processing.

The authors searched standard literature databases using a string of keywords to identify relevant articles and then excluded articles based on several criteria, including excluding non-western languages, age, and the presence of specific disorders. This methodology has inclusion/exclusion criteria clearly stated and allows some replicability of their findings (although the final selection process also involved two external consultants and some internal deliberation).

The authors then focus on twelve selected articles to assess the association between music and language abilities in children: they observe that musical skills predict children's performance on language tasks, including semantic and syntactic aspects as well as phonological awareness; music and language share similarities in the cognitive and neural mechanisms active in the acquisition and/or processing of music and language. The authors also point out that rhythmic components of music and language are predictive of infants' language development.

For such an ambitious and complex task, the authors have undertaken a fair review on the current understanding of the topic. However, the paper must be improved by addressing the following concerns:

1.       Oversimplification: (1) The paper claims that language acquisition in children is fast and largely autonomous, while music is acquired more slowly and depends on substantial teaching and practice. While there is some truth to this claim, it is also an oversimplification, as both language and music acquisition involve complex interactions between genetic, cognitive, and environmental factors. These other factors must be included in the discussion. (2) The authors tend to oversimplify the relationship between music and language by suggesting that they are essentially the same thing. While there are certainly many similarities between music and language, they are also distinct and separate systems with their own unique features and functions. (3) Another oversimplification is the suggestion that music has a privileged status in the infant's brain, but the concept of the privileged status of music is precisely still debated in the field.

2.       Excessive Exclusion: (1) The deliberate exclusion of articles focusing on music therapy or music rehabilitation might have limited the scope of the review, as understanding disorders often also provide fundamental insight into the nature of the phenomena studied. Perhaps address this limitation in the discussion? (2) Excluding articles investigating the effect of musical training on language development could also limit the review's findings. Perhaps address such limitation in the discussion?

3.       Poor representation of non-western languages: the review's focus on studies in English, German and largely ‘western’ languages (granted, including Turkish) could introduce a bias towards these languages and limit the generalizability of the findings to other languages. For example, no tonal languages are included – the tonal aspect may offer crucial insight connecting language and music. Such limitation should be acknowledged in the discussion.

4.       Numerous standardized tests (e.g. to assess musical ability) are referenced in the text, but insufficient information about these tests are given to the reader; could a summary be provided as an appendix?

5.       Paragraph at Lines 275-290 is written poorly, with much repetition. Bizarrely, the final sentence at Line 289-290 seems incomplete!

6.       Similarly, the following Paragraph (Lines 291-299) is similarly garbled in writing, with truncated sentences and weird punctuation (very painful to read). Rewrite! Grammar is also problematic – please seek help from a professional editor or native speaker of English.

7.       Table 1: Currently very poor readability. Perhaps present the table as Landscape (90% rotated) to take advantage of wider column space, so that text does not keep breaking across lines.

8.       Missing Discussion: The Authors focus on the similarities between music and language, but do not discuss any differences or limitations of the shared mechanisms. The text could also benefit from more discussion on the implications of the research, such as how music can be used to promote language development.

9.       Missing Conclusion Section! The review is otherwise incomplete and betrays shoddy scholarship to an otherwise insightful article.

Other Minor comments:

1.       Lines 71-74: Weird sentence structure with questionable punctuation (e.g.
“… whereas adults, process language…”) and grammar. Check and re-write.

2.       Line 126: “Disagreements were resolved through discussion” is a curious statement to report – how else ought disagreements be resolved? Isn’t discussion an expected activity amongst co-authors? How does this activity warrant reporting?

3.       Line 146 (and elsewhere) “about 66% of articles” when there are only 12 articles, i.e. 8 articles. There is no “about 66%” (either there is 66% or there isn’t). Instead of percentage for only 12 articles, why not just state the number, e.g. 8 of the 12 articles, or 3 of the 12 articles, etc. Instead of 58% (line 149) why not just simply say 7 articles? Instead of 16% (line 150) say 2 articles? Also Line 201 (“About 58%”), etc.

4.       Line 296 “…textural information…” – what does this mean? Give a definition. Or did the authors mean “textual information”?

5.       Line 321 “gesture development” – what does this mean? Give a definition. This term is used without prior explanation.

6.       Line 327 “Both these findings [33,35], represent the…” redundant comma – remove!

7.       Line 346 “The rhythm…” No need article “the” here, as you are describing a general phenomenon instead of something specific.

8.       Line 350 “…language belonging to the same rhythmic class…” How is a language’s rhythmic class established? Cite reference for this sentence, not just the next sentence.

9.       Line 390 “…spoken words, sung words and vocalize.” Do you mean instead “vocalese”?

10.   Grammar: Line 406 “Our decision does not include these studies…” do you mean instead “Our decision to not include these studies…”

11.   Grammar: Line 409 “…durations and frequencies, settings.” Weird way to list things in a sentence. Rewrite!

12.   Non-English Sentence Structure: Line 409-410

13.   Non-English Structure: Line 411. Do you instead mean “Despite the limitations described above…”

Author Response

Review 1

This paper provides a limited review of the relationship between music and language in children, focusing on their similarities and differences, the processes involved in their acquisition, and touches vaguely the neural mechanisms underlying their processing.

 The authors searched standard literature databases using a string of keywords to identify relevant articles and then excluded articles based on several criteria, including excluding non-western languages, age, and the presence of specific disorders. This methodology has inclusion/exclusion criteria clearly stated and allows some replicability of their findings (although the final selection process also involved two external consultants and some internal deliberation).

R:We want to emphasize that we have not deliberately excluded non-western languages, it is not an excluding criterion. We explain in detail below

The authors then focus on twelve selected articles to assess the association between music and language abilities in children: they observe that musical skills predict children's performance on language tasks, including semantic and syntactic aspects as well as phonological awareness; music and language share similarities in the cognitive and neural mechanisms active in the acquisition and/or processing of music and language. The authors also point out that rhythmic components of music and language are predictive of infants' language development.

For such an ambitious and complex task, the authors have undertaken a fair review on the current understanding of the topic. However, the paper must be improved by addressing the following concerns:

  1. Oversimplification: (1) The paper claims that language acquisition in children is fast and largely autonomous, while music is acquired more slowly and depends on substantial teaching and practice. While there is some truth to this claim, it is also an oversimplification, as both language and music acquisition involve complex interactions between genetic, cognitive, and environmental factors. These other factors must be included in the discussion. (2) The authors tend to oversimplify the relationship between music and language by suggesting that they are essentially the same thing. While there are certainly many similarities between music and language, they are also distinct and separate systems with their own unique features and functions. (3) Another oversimplification is the suggestion that music has a privileged status in the infant's brain, but the concept of the privileged status of music is precisely still debated in the field.

R:We agree with the reviewer, thus we modified the introduction and discussion sections. We think that the introduction section is now improved along with the reviewer’s suggestion.

Specifically regarding suggestions:

  • You can see the correction in Lines 98-100 and in lines 423-427.
  • You can see the correction in Lines 87-97.
  • You can see the correction in Lines 393-394.

  1. Excessive Exclusion: (1) The deliberate exclusion of articles focusing on music therapy or music rehabilitation might have limited the scope of the review, as understanding disorders often also provide fundamental insight into the nature of the phenomena studied. Perhaps address this limitation in the discussion? (2) Excluding articles investigating the effect of musical training on language development could also limit the review's findings. Perhaps address such limitation in the discussion?

R:Thank you for your suggestion, we have included in the discussion section a statement on the limitation of this study regarding the exclusion of articles focusing on music interventions (music training, music therapy, and music rehabilitation). You can see the paragraph in Lines 414-421. We explained in detail the motivation for our decision recognizing that this might have limited the scope of the review. Our future aim is to investigate the specific role of music interventions on language development by analyzing all variables included in these types of interventions, such as approaches and modalities of music, individual or group levels, professional figures involved, frequency and duration of the interventions, and settings (e.g., at home or school).

  1. Poor representation of non-western languages: the review's focus on studies in English, German and largely ‘western’ languages (granted, including Turkish) could introduce a bias towards these languages and limit the generalizability of the findings to other languages. For example, no tonal languages are included – the tonal aspect may offer crucial insight connecting language and music. Such limitation should be acknowledged in the discussion.

R:The reviewer's suggestion is interesting because he captures the lack of studies on non-western languages. Although we did not deliberately exclude papers on Eastern languages during the search process, to our knowledge, no studies directly investigated the association between music and language according to our exclusion/inclusion criteria. Specifically, you can see below some examples of articles that emerged during the study selection process. In the next step, these articles were excluded by authors  because they aren't considered eligible, based on inclusion and exclusion criteria:

Chen-Hafteck, L., Niekerk, C.V., Lebaka, E., & Masuelele, P. (1999). Effects of Language Characteristics on Children's Singing Pitch: Some Observations on Sotho- and English-speaking Children's Singing. Bulletin of the Council for Research in Music Education.

Guo, H., Yuan, W., Fung, C. V., Chen, F., & Li, Y. (2022). The relationship between extracurricular music activity participation and music and Chinese language academic achievements of primary school students in China. Psychology of Music, 50(3), 742–           755. https://doi.org/10.1177/03057356211027642

Gen’ichi, T. (2004). Coercively Standardized or Not: Romanization Systems of the Japanese Language in Music Literature. The World of Music, 46(2), 137–143. http://www.jstor.org/stable/41699570

Zhao TC, Llanos F, Chandrasekaran B, Kuhl PK. Language experience during the sensitive period narrows infants' sensory encoding of lexical tones-Music intervention reverses it. Front Hum Neurosci. 2022 Jul 9;16:941853. doi: 10.3389/fnhum.2022.941853. PMID: 36016666; PMCID: PMC9398460.

We believe that this is not a limitation of our review, but rather a general limitation of research in this specific field. If the reviewer is aware of any research that he or she believes is appropriate for our objective, we would ask him or her to suggest it to us.

  1. Numerous standardized tests (e.g. to assess musical ability) are referenced in the text, but insufficient information about these tests are given to the reader; could a summary be provided as an appendix?

R:Thank you for your useful suggestion. We have added Appendix A (music measures) and Appendix B (language measures).

  1. Paragraph at Lines 275-290 is written poorly, with much repetition. Bizarrely, the final sentence at Line 289-290 seems incomplete!

R:Thank you for your suggestion. We have modified the sentence as requested by the reviewer.

  1. Similarly, the following Paragraph (Lines 291-299) is similarly garbled in writing, with truncated sentences and weird punctuation (very painful to read). Rewrite! Grammar is also problematic – please seek help from a professional editor or native speaker of English.

R:Thank you for your suggestion. We have modified the sentence as requested by the reviewer.

  1. Table 1: Currently very poor readability. Perhaps present the table as Landscape (90% rotated) to take advantage of wider column space, so that text does not keep breaking across lines.

R:Thank you for your suggestion. We hope that the table is readable now.

  1. Missing Discussion: The Authors focus on the similarities between music and language, but do not discuss any differences or limitations of the shared mechanisms. The text could also benefit from more discussion on the implications of the research, such as how music can be used to promote language development.

R:Thank you for your suggestions. We added the implication of our review section in discussion section.

  1. Missing Conclusion Section! The review is otherwise incomplete and betrays shoddy scholarship to an otherwise insightful article.

R:Thank you for the suggestion. We added the conclusion section in our review.

Other Minor comments:

  1. Lines 71-74: Weird sentence structure with questionable punctuation (e.g.
    “… whereas adults, process language…”) and grammar. Check and re-write.

R:Thank you for the suggestion. We have completely rewritten the sentence.

  1. Line 126: “Disagreements were resolved through discussion” is a curious statement to report – how else ought disagreements be resolved? Isn’t discussion an expected activity amongst co-authors? How does this activity warrant reporting?

R:You can see the correction in line 146.

  1. Line 146 (and elsewhere) “about66% of articles” when there are only 12 articles, i.e. 8 articles. There is no “about 66%” (either there is 66% or there isn’t). Instead of percentage for only 12 articles, why not just state the number, e.g. 8 of the 12 articles, or 3 of the 12 articles, etc. Instead of 58% (line 149) why not just simply say 7 articles? Instead of 16% (line 150) say 2 articles? Also Line 201 (“About 58%”), etc.

R:The reviewer is correct. We changed the percentage to an exact number.

  1. Line 296 “…textural information…” – what does this mean? Give a definition. Or did the authors mean “textual information”?

R:Thank you to catch a mistake. We have modified textual information with the textural sound sequence as reported in the article of Canette et al. [42].

  1. Line 321 “gesture development” – what does this mean? Give a definition. This term is used without prior explanation.

R:We have explained the meaning of gesture development. You can see lines 299-302.

  1. Line 327 “Both these findings [33,35], represent the…” redundant comma – remove!

R:Thank ok for your suggestion. We removed the redundant comma.

  1. Line 346 “The rhythm…” No need article “the” here, as you are describing a general phenomenon instead of something specific.

R:Thank you for your suggestion. We have to delete the article “the” in the sentence as requested by      the reviewer.

  1. Line 350 “…language belonging to the same rhythmic class…” How is a language’s rhythmic class established? Cite reference for this sentence, not just the next sentence.

R:You can see the modified sentence in lines 341-353.

  1. Line 390 “…spoken words, sung words and vocalize.” Do you mean instead “vocalese”?

R:We explained “vocalise” term as suggested by the original paper [92] in lines 392-393.

  1. Grammar: Line 406 “Our decision does not include these studies…” do you mean instead “Our decision tonot include these studies…”

R:Thank you to catch a mistake.

  1. Grammar: Line 409 “…durations and frequencies, settings.” Weird way to list things in a sentence. Rewrite!
  2. Non-English Sentence Structure: Line 409-410
  3. Non-English Structure: Line 411. Do you instead mean “Despite the limitations described above…”

R:For the 11, 12, and 13 comments: we revised the English in detail and corrected errors in the manuscript.

Finally, we would like to thank the reviewer for his comments because we feel that thanks to him our       review has improved considerably.

Reviewer 2 Report

This is a review about the relationship between music and language, which highlights the fact that music plays a critical role in language development in early life. It is an interesting study, being of interest to the general population. It is correctly structured, the methods by which it was achieved being well described.
Line 114...please explain in a parenthesis what L2 means.
Line 294...please correct/finish the sentence "In the semantic fluency task, where children had to produce list of words, authors found a specific"
I recommend adding a separate paragraph with the conclusions.
The references are appropriate, the article presents 84 references, being up to date. Please review the writing of the references, some are not complete - the number of pages is missing, for example. Also, in some references the year is written in bold letters and in others not.

Author Response

Review 2

This is a review about the relationship between music and language, which highlights the fact that music plays a critical role in language development in early life. It is an interesting study, being of            interest to the general population. It is correctly structured, the methods by which it was achieved        being well described.

R:Thank you for your positive comment regarding our review.

Line 114...please explain in a parenthesis what L2 means.

R:You can see the correction in line 134.

Line 294...please correct/finish the sentence "In the semantic fluency task, where children had to produce list of words, authors found a specific"

R:Thank you for the suggestion. We completely modified the sentence.

I recommend adding a separate paragraph with the conclusions.

R:Thank you for the suggestion. We added the conclusion section in our review.

The references are appropriate, the article presents 84 references, being up to date. Please review       the writing of the references, some are not complete - the number of pages is missing, for example. Also, in some references the year is written in bold letters and in others not.

R: The reviewer is correct. We reviewed the references

Round 2

Reviewer 1 Report

The authors have satisfactorily addressed my points raised. However three issues now arise:

1. Poor representation of non-western languages: The Authors point out (rightly so) “that this is not a limitation of our review, but rather a general limitation of research in this specific field”. An excellent point which deserves specific mention in the Discussion as one of the limitations not of the review but a matter-of-fact that current studies in the field poorly represent non-western languages, which could introduce bias and limit the generalizability of the findings to other languages.

2. Table 1: Indeed more readable now. However, this precisely now exposes poor writing in the summary text provided in the table. In the current form, Table 1 is sloppy! For example,

-          Inconsistent [missing] spacing (e.g. Ref [1] “SDage= 4.7 months”; missing spacing after bullet points “-” in many places)

-          Ref [15] has missing numbering for Questionnaire 2

-          Ref [26] has mysterious extra numbers “76” in the text “Bayley Scales of Infant Development76 (BSID-III; [53])

-          Ref [29] has random line-breaks (starts new lines mid-sentence); also has missing fullstop at end of sentence “musical pitch, one of the building blocks of music

-          Ref [30] has missing hyphen in the text “Cross sectional” when other Ref e.g. [15] and [29] include hyphen in the word “Cross-sectional

-          Ref [32] has strange use of semi-colon in the text “25 English children (13M;12F), aged 5;11 to 7;1 years (M = 6;6 years, SD = 4 months)”; do the authors instead wish to use decimal points like in the next Ref [33]? Alternatively, if Year;Month is referred, entry for Ref [42] spells this out in full without ambiguity, and is much preferred. To make Table 1 meaningful for comparing across studies, please be consistent!

-          Similarly, Ref [41] has strange use of semi-colon in the text “16 English children aged 5;6 to 8;7 (M = 6;5, SD = 11 months; 9M, 7F)

Please check all the entries in Table 1 carefully for consistent formatting, punctuation, abbreviations, random (missing?) spaces, bullet points, page-breaks/words suddenly starting on new line mid-sentence, etc.

3. Line 393 “Thus, it has been hypnotized that music…” – did the authors mean instead “Thus, it has been hypothesized that music…”?

Author Response

Dear Reviewer,

we feel revisions have strengthened the manuscript and appreciate your feedback.

Best regards, 

S.D'Amico

The authors have satisfactorily addressed my points raised. However, three issues now arise:

  1. Poor representation of non-western languages: The Authors point out (rightly so) “that this is not a limitation of our review, but rather a general limitation of research in this specific field”. An excellent point which deserves specific mention in the Discussion as one of the limitations not of the review but a matter-of-fact that current studies in the field poorly represent non-western languages, which could introduce bias and limit the generalizability of the findings to other languages.

R. We added this aspect in the discussion section as requested by the reviewer. You can see the sentence in linea 423-428.

  1. Table 1: Indeed more readable now. However, this precisely now exposes poor writing in the summary text provided in the table. In the current form, Table 1 is sloppy! For example,

-          Inconsistent [missing] spacing (e.g. Ref [1] “SDage= 4.7 months”; missing spacing after bullet points “-” in many places)

-          Ref [15] has missing numbering for Questionnaire 2

-          Ref [26] has mysterious extra numbers “76” in the text “Bayley Scales of Infant Development76 (BSID-III; [53])”

-          Ref [29] has random line-breaks (starts new lines mid-sentence); also has missing fullstop at end of sentence “musical pitch, one of the building blocks of music ”

-          Ref [30] has missing hyphen in the text “Cross sectional” when other Ref e.g. [15] and [29] include hyphen in the word “Cross-sectional”

-          Ref [32] has strange use of semi-colon in the text “25 English children (13M;12F), aged 5;11 to 7;1 years (M = 6;6 years, SD = 4 months)”; do the authors instead wish to use decimal points like in the next Ref [33]? Alternatively, if Year;Month is referred, entry for Ref [42] spells this out in full without ambiguity, and is much preferred. To make Table 1 meaningful for comparing across studies, please be consistent!

-          Similarly, Ref [41] has strange use of semi-colon in the text “16 English children aged 5;6 to 8;7 (M = 6;5, SD = 11 months; 9M, 7F)”

Please check all the entries in Table 1 carefully for consistent formatting, punctuation, abbreviations, random (missing?) spaces, bullet points, page-breaks/words suddenly starting on new line mid-sentence, etc.

R.Yes, it’s a mistake, we thank the reviewer for capturing this error. 

  1. Line 393 “Thus, it has been hypnotized that music…” – did the authors mean instead “Thus, it has been hypothesized that music…”?

 R.Yes, it’s a mistake, we thank the reviewer for capturing this error.